

# PARROT: ENHANCING MULTI-TURN CHAT MODELS BY LEARNING TO ASK QUESTIONS

## ABSTRACT

Impressive progress has been made on chat models based on Large Language Models (LLMs) recently; however, there is a noticeable lag in multi-turn conversations between open-source chat models (e.g., Alpaca and Vicuna) and the leading chat models (e.g., ChatGPT and GPT-4). Through a series of analyses, we attribute the lag to the lack of enough high-quality multi-turn instruction-tuning data. The available instruction-tuning data for the community are either single-turn conversations or multi-turn ones with certain issues, such as non-human-like instructions, less detailed responses, or rare topic shifts. In this paper, we address these challenges by introducing Parrot, a highly scalable solution designed to automatically generate high-quality instruction-tuning data, which are then used to enhance the effectiveness of chat models in multi-turn conversations. Specifically, we start by training the Parrot-Ask model, which is designed to emulate real users in generating instructions. We then utilize Parrot-Ask to engage in multi-turn conversations with ChatGPT across a diverse range of topics, resulting in a collection of 40K high-quality multi-turn dialogues (Parrot-40K). These data are subsequently employed to train a chat model that we have named Parrot-Chat. We demonstrate that the dialogues gathered from Parrot-Ask markedly outperform existing multi-turn instruction-following datasets in critical metrics, including topic diversity, number of turns, and resemblance to human conversation. With only 40K training examples, Parrot-Chat achieves strong performance against other 13B open-source models across a range of instruction-following benchmarks, and particularly excels in evaluations of multi-turn capabilities. All codes and datasets will be publicly available to facilitate future research.

## 1 INTRODUCTION

Recently, large language models (LLMs) based chat models (OpenAI, 2022; 2023; Taori et al., 2023; Chiang et al., 2023; Xu et al., 2023; Ding et al., 2023) have demonstrated their strong capability in understanding human instructions and generating helpful responses. A series of open-source chat models fine-tuned from the LLaMA model (Touvron et al., 2023a;b) show promising results, even in some benchmarks performing closely to ChatGPT even GPT-4 (Chiang et al., 2023). However, in multi-turn conversations, their instruction-following performance seems to degrade as the number of dialogue turns increases (Zheng et al., 2023). Our quantitative analysis also indicates that for many open-source chat models, there is an obvious drop in performance when evaluated on dialogues with more than six turns (More details in Sec. 4.3).

To train a chat model that is well-aligned with humans, a key challenge is to construct a high-quality instruction-response dataset. Leading performing chat models like ChatGPT and GPT-4 employ annotators to iteratively craft high-quality human instructions and responses. As manual annotation is time-consuming and labor-intensive, the open-source community turns to investigate methods that automatically generate multi-turn instructions and responses from ChatGPT or GPT-4 (Xu et al., 2023; Ding et al., 2023). These methods generally fall into two categories: self-chatting and iterative self-chatting. The self-chatting paradigm (Xu et al., 2023) generates dialogues by prompting ChatGPT with dialogue generation instructions, while the iterative self-chatting paradigm (Ding et al., 2023) employs two ChatGPT APIs to conduct multi-turn chats. Fig. 1 illustrates the two

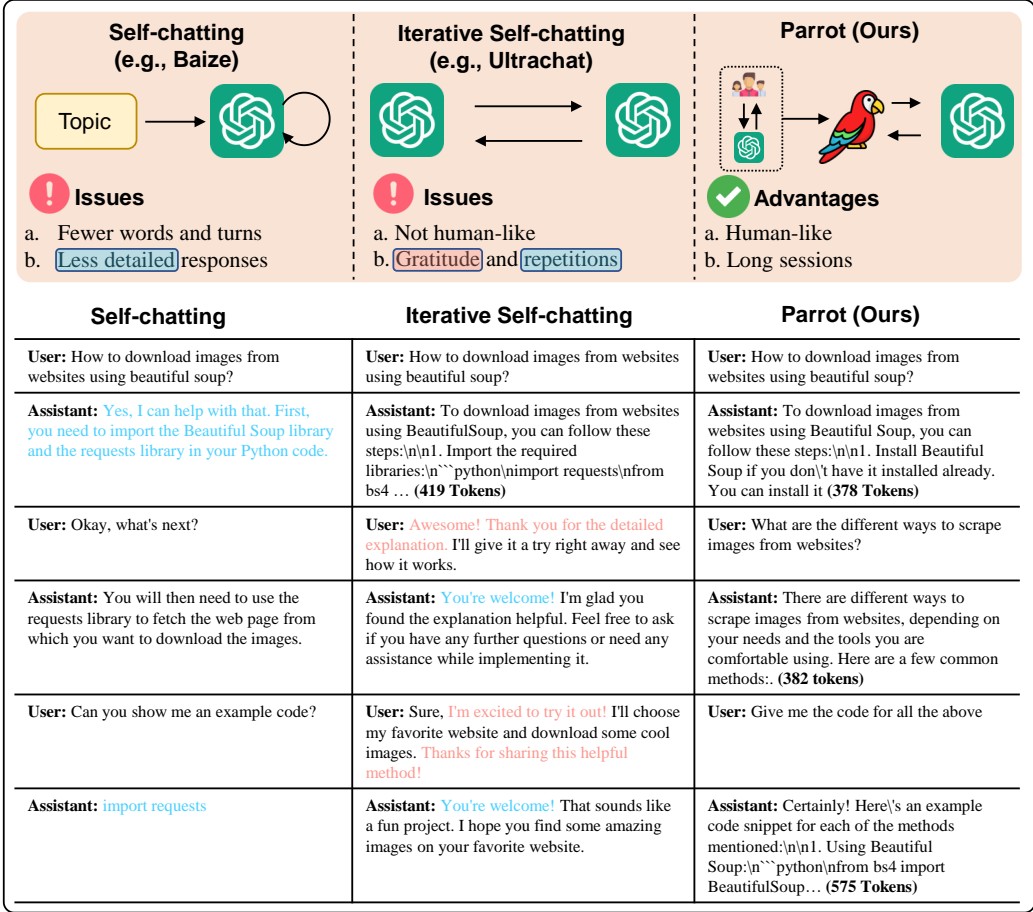

Figure 1: Comparison of different methods of collecting multi-turn instruction-tuning data. All of these data have certain issues that may lead to poor performance on multi-turn chat. Self-chatting tends to generate less detailed responses, while iterative self-chatting often leads to unnecessary repetitions or expressions of gratitude.

paradigms, each with dialogue as an example. As shown in the figure, both the two paradigms have certain issues. The self-chatting paradigm usually results in less detailed responses, while the iterative self-chatting paradigm tends to generate unnecessary repetitions or expressions of gratitude. Additionally, instructions generated by automatic approaches are not human-like: they lack colloquial expression, are usually long and not specific, and rarely exhibit topic shifts and transitions. This makes it challenging to construct meaningful long-turn dialogues, which is crucial for unleashing the instruction-following capability for LLMs. There are also works (Chiang et al., 2023) that utilize the available user-ChatGPT logs from online platforms such as ShareGPT to fine-tune LLMs as chat models. However, such logs are typically scarce and tend to be biased towards shorter sessions, which are still insufficient to achieve satisfying results.

To overcome the above limitations, we introduce a novel paradigm called Parrot, a highly scalable approach designed for automatically collecting high-quality instruction-tuning data and enhancing the multi-turn instruction-following capabilities of chat models. Different from previous works (Taori et al., 2023; Chiang et al., 2023; Xu et al., 2023; Ding et al., 2023) that focus on training LLMs to respond to users, we propose to train LLMs to learn to ask human-like questions. We name the asking model as Parrot-Ask. Then we leverage Parrot-Ask to chat with ChatGPT in multi-rounds under a wide range of topics. In this way, we collect 40K high-quality multi-turn conversations (Parrot-40K). Finally, these data are used to fine-tune an LLM as a chat model, which is named Parrot-Chat.

We conduct extensive experiments to verify the effectiveness of our approach. Compared with previous approaches, our Parrot-Ask model generates higher-quality data with more human-style instructions. Parrot-Ask rarely exhibits non-human-like behavior when asking questions, such as posing non-specific questions, repeating similar instructions, and expressing excessive gratitude. Such characteristics make Parrot-40K filled with more meaningful conversations, reaching an average turn of 8.7, which far exceeds existing datasets. Fine-tuning on Parrot-40K, our Parrot-Chat model achieves superior performance among open-source models across a range of instruction-following benchmarks, including MT-Bench (Zheng et al., 2023) and Alpaca-eval (Li et al., 2023b), especially on our constructed multi-turn evaluation benchmark MT-Bench++. We also conduct ablation studies to verify the effectiveness of each component in our approach. Through these ablation studies, we demonstrate that training data quality and the number of training dialogue turns both play crucial roles in enhancing the multi-turn performance of the chat model.

We summarize our main contributions as follows:

- We systematically explore the multi-turn instruction-following capabilities of chat models. Through analysis, we show that a high-quality instruction-tuning dataset plays a key role in empowering the multi-turn instruction-following capabilities of the chat models.
- We propose Parrot, a scalable paradigm to automatically construct human-like instruction-following conversations, which can be used to enhance the performance of chat models.
- We collect Parrot-40K using Parrot-Ask, which is superior to existing datasets in numerous critical metrics, including topic diversity, number of turns, and resemblance to human conversation.
- With only 40K training dialogues, our Parrot-Chat model achieves superior performance among 13B open-source chat models across a range of instruction-following benchmarks and particularly excels in evaluations of multi-turn capabilities.

## 2 RELATED WORK

### 2.1 LLMS FOR CHAT

Pretrained Language Models have developed rapidly in recent years (OpenAI, 2022; Ouyang et al., 2022; OpenAI, 2023; Brown et al., 2020; Radford et al., 2019; Chowdhery et al., 2022; Devlin et al., 2019). Due to their strong language understanding and generation capabilities, these models are also adopted in dialogue systems. Earlier works utilize BERT-style models for retrieval-based dialogue models (Han et al., 2021; Gu et al., 2020), or directly fine-tune a GPT-style model on conversation data (Shuster et al., 2022; Zhang et al., 2020; Bao et al., 2020). However, due to the limited capabilities of their foundation models, these models have not been large-scale used in real life. GPT-3 (Brown et al., 2020) demonstrates that language generation ability can be largely improved by scaling model parameters and pre-training data. However, it is not aligned with humans and cannot understand user instructions well. Instruction tuning is then proposed to align language models to human instructions (Ouyang et al., 2022; Wei et al., 2021; Wang et al., 2022b). Some works focus on building instruction-tuning data for existing NLP tasks, the models trained on these data show powerful zero-shot generalization on unseen tasks (Wei et al., 2021; Wang et al., 2022b). Instruct-GPT (Ouyang et al., 2022) collects real user queries and uses human-annotated responses as training data, and thus more aligned with real user requirements and has better generalization ability. It also adopts Reinforcement Learning from Human Feedback to further align itself with humans. Chat-GPT (OpenAI, 2022) and GPT4 (OpenAI, 2023) expand this paradigm to multi-turn scenarios and lead to an effective LLM-based chat model. The open-source community also leverages instruction tuning to build chat models based on open-source LLMs (Taori et al., 2023; Chiang et al., 2023; Ding et al., 2023; Xu et al., 2023). Different from these models that serve to give answers, we aim to train a model to ask questions under the given context, which can be used to collect high-quality multi-turn instruction-tuning data.

### 2.2 INSTRUCTION GENERATION WITH LLMS

Instruction tuning plays an important role in inspiring the instruction-following ability of LLMs and aligning with humans (Wang et al., 2022b; Wei et al., 2021; Ouyang et al., 2022; OpenAI,

2022; 2023). Due to the expensive costs to collect human-annotated instruction tuning data such as ChatGPT (OpenAI, 2022) and GPT4 (OpenAI, 2023), recent works explore leveraging the powerful LLMs to generate instruction-response pairs in an automatic manner (Taori et al., 2023; Ding et al., 2023; Xu et al., 2023; Wang et al., 2022a). Self-Instruct (Wang et al., 2022a) is a prior work that uses GPT-3 to generate instructions. It designs seed prompts as examples to prompt GPT-3 to generate more instructions. It also designs a filter process based on ROUGE-L to ensure the diversity of generated instructions. Alpaca (Taori et al., 2023) adopts the same pipeline to collect instruction-response pairs using ChatGPT and then fine-tuning a LLaMA model (Touvron et al., 2023a). A recent work Humpback (Li et al., 2023a) proposes instruction back-translation that trains an LLM to generate instructions for web corpus. However, they mainly focus on single-turn instructions. Later works follow this paradigm and enhance the performance by further improving the diversity and amount of instruction data, especially including multi-turn instructions for tuning a chat model (Xu et al., 2023; Ding et al., 2023). Baize (Xu et al., 2023) collects multi-turn instructions by leveraging ChatGPT in a self-chatting manner. UltraChat (Ding et al., 2023) utilize two ChatGPT API to play the role of user and assistant separately and improves the amount and quality of instructional conversational data. Vicuna (Chiang et al., 2023) adopts user-ChatGPT logs from the ShareGPT platform for training chat models. However, these multi-turn instructional conversational data still have several drawbacks, such as less detailed responses, not human-like instruction, or a limited number of sessions. In this paper, we propose the Parrot approach, where we train the Parrot-Ask model to ask multi-turn questions and engage in conversations with ChatGPT, to automatically construct high-quality multi-turn instruction dialogues.

## 2.3 EVALUATIONS OF CHAT MODELS

The current benchmarks for LLM-based chat models are mainly focused on single-turn evaluation (Hendrycks et al., 2020; Zhong et al., 2023; Srivastava et al., 2022; Li et al., 2023b). MMLU (Hendrycks et al., 2020) and Big-bench (Srivastava et al., 2022) are designed as multiple-choice questions to measure the knowledge and reasoning ability of LLMs. AGIEval (Zhong et al., 2023) constructs human-centric evaluations for LLMs from standardized exams. However, these kinds of evaluations obey the nature of the open-ended generation of chat models and cannot reflect the ability to follow user instructions (Zheng et al., 2023). Alpaca-eval (Li et al., 2023b) builds a single-turn instruction following benchmark with 805 open-ended questions and adopts strong LLMs such as GPT-4 to give evaluation. Chatbot Arena (Zheng et al., 2023) is a platform where users can vote to compare diverse chat models. MT-Bench (Zheng et al., 2023) builds the first evaluation benchmarks for multi-turn instruction-following, it simulates real dialogue scenarios by designing questions in 8 categories, including writing, coding, reasoning, etc. It adopts GPT-4 to judge the quality of model answers and shows there is a high agreement to human evaluation. However, MT-Bench only contains two questions for each session, thus cannot reflect the ability of chat models to handle multi-turn questions. In this paper, we construct an 8-question evaluation benchmark based on MT-Bench to evaluate the multi-turn ability of chat models.

## 3 APPROACH

In this section, we describe our proposed framework called Parrot. As illustrated in Fig. 2, we first train a Parrot-Ask model to mimic the asking style of humans in generating multi-turn instructions. Then we use our Parrot-Ask model to chat with ChatGPT under diverse topics to automatically collect a high-quality multi-turn instruction-tuning dataset called Parrot-40K. Finally, we use the collected dataset to train a Parrot-Chat model.

### 3.1 LEARNING TO ASK QUESTIONS

Human dialogues encompass a wide array of styles and behaviors. For instance, humans often employ short spoken language filled with coreferences, and instructions, typically specific, tend to diverge toward related topics as the conversation progresses. However, as ChatGPT is designed to function as an AI assistant, emulating these characteristics can be inherently challenging, even when we prompt with complex instructions to simulate a user. To overcome this challenge, we train the specialized model to pose questions, called Parrot-Ask, based on LLaMA (Touvron et al., 2023a) using the available real user-ChatGPT logs. Leveraging the capabilities of powerful LLMs, we have

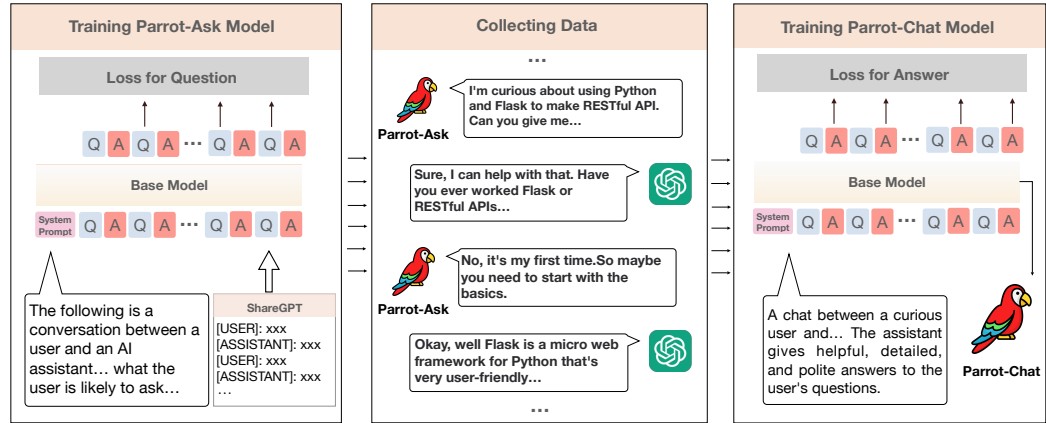

Figure 2: The overall framework of Parrot. (a) First, we train the Parrot-Ask model on real user-ChatGPT logs to learn how real users pose questions. (b) We then utilize the trained question-asking model to iteratively chatting with ChatGPT across a variety of topics, thereby collecting high-quality multi-turn instruction-response pairs. (c) Finally, we use the collected data to train the ultimate Parrot-Chat model, thereby enhancing its multi-turn performance.

successfully developed an asking model using only tens of thousands of training examples, which is capable of generating specific and meaningful questions with natural topic shifts (see Sec. 4.2).

The process of training the Parrot-Ask model is essentially the inverse of training a chat model. Specifically, chat models like Vicuna are trained by predicting answer tokens, conditioned on the user's question and the conversation history; whereas, Parrot-Ask model are trained to predict question tokens, conditioned on the assistant's answer and the history. Accordingly, we modify the training loss to focus exclusively on the user's question tokens. This adjustment enables the model to learn to generate questions.

For a multi-turn instructional conversation consisting of $T$ question-response pairs $X = (X_q^1, X_a^1, X_q^2, X_a^2, ..., X_q^T, X_a^T,)$, we compute the loss on the predicted question tokens using auto-regressive training objective as:

$$\mathcal{L}_{ask} = \sum_i^L \log p(x_i | X_{q,<i}, X_{a,<i}), x_i \in X_q, \tag{1}$$

where $L$ is the token length of sequence $X$, $x_i$ is the current predicted question tokens, $X_{q,<i}$ and $X_{a,<i}$ are the context questions and answers tokens before $x_i$, respectively. It is worth noting that we treat the initial round as a part of the context and do not calculate the loss on it. In our experiments, the first question comes from real users to make sure the topic has true values to humans. We leave the trial to generate the first question as future work.

### 3.2 COLLECTING MULTI-TURN INSTRUCTION-TUNING DATA

Similar to recent methods such as the self-instruct (Wang et al., 2022a) and the iterative self-chatting paradigm (Xu et al., 2023; Ding et al., 2023), we utilize gpt-3.5-turbo to produce responses corresponding to the generated questions or instructions. Here we create multi-turn instruction-response pairs in two parts: 1) we expand the existing sessions in the ShareGPT dataset to be longer. Thus we can leverage the high-quality existing data. Given an existing sequence of $X = (X_q^1, X_a^1, ..., X_q^N, X_a^N)$, we utilize Parrot-Ask to generate a new follow-up question denoted as $X_q^{N+1}$. Subsequently, we use gpt-3.5-turbo to generate an answer $X_a^{N+1}$. This process is repeated iteratively until we reach the desired number of conversational turns or gpt-3.5-turbo returns meaningless answers. 2) we take the first instructions in the session of UltraChat (Ding et al., 2023) as initial questions and generate long sessions. Thus we can leverage the topic coverage and diversity of UltraChat. Given an initial question $X_q^1$, we first use gpt-3.5-turbo to generate an appropriate answer $X_a^1$. We then employ Parrot-Ask to generate a new question, denoted as $X_a^2$. This conversa-

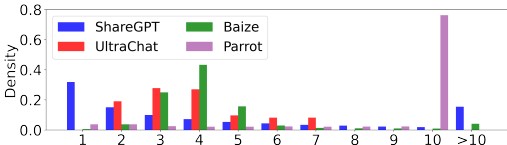

| Dataset | #Avg. Turns | #Tokens | #Topics | #RefCount |
|---------|------------|---------|---------|-----------|
| Baize | 4.54 | 269 | 3.1 | 5.0 |
| UltraChat | 3.85 | 1455 | 2.3 | 9.9 |
| ShareGPT | 6.67 | 3166 | 3.3 | 11.1 |
| Parrot | **8.71** | **3551** | **5.5** | **11.6** |

Figure 3: Distribution of the number of dialogue turns in each training dataset.

Table 1: Statistics of Parrot dataset and comparison with other multi-turn instruction-tuning datasets.

tion is then continued by iteratively generating subsequent questions and answers until we reach the target number of conversational turns or gpt-3.5-turbo returns meaningless answers.

### 3.3 TRAINING THE CHAT MODEL

The training of Parrot-Chat is consistent with other open-source chat models(Taori et al., 2023; Chiang et al., 2023). With the system instruction and user question tokens masked, we compute the loss on the response tokens as:

$$\mathcal{L}_{assistant} = \sum_{i}^{L} \log p(x_i | X_{q,<i}, X_{a,<i}), x_i \in X_a. \qquad (2)$$

where $L$ is the token length of sequence $X$, $x_i$ is the current predicted answer tokens, $X_{q,<i}$ and $X_{a,<i}$ are the context questions and answers tokens before $x_i$.

## 4 EXPERIMENT

In this section, we conduct extensive experiments to verify the effectiveness of our Parrot-Ask and Parrot-Chat models. We first compare our constructed Parrot-40K dataset with existing multi-turn instruction-following datasets. As will be shown, Parrot-40K is superior on metrics such as the valid number of turns, topic diversity, and human-likeness against all datasets collected from other approaches, and surpasses its training dataset ShareGPT. Then, to adequately assess the multi-turn capabilities of the chat models, we construct MT-Bench++ evaluation benchmark by extending each dialogue in MT-Bench with six manually written instructions. We also show that most open-source chat models have a performance drop when they are evaluated on dialogues of more than six turns. We further evaluate Parrot-Chat across multiple benchmarks and compare it with the baseline models. Finally, through ablation studies, we underscore the importance of data quality and session length in enhancing the capability of the chat models to handle the multi-turn instruction-following task.

### 4.1 TRAINING DETAILS

**Parrot-Ask Training Details.** We build our Parrot-Ask model on the open-source LLaMa2-13B model. We train it using 70K ShareGPT sessions. We adopt a max length of 4096 tokens for the model. We train the model for 3 epochs with AdamW optimizer in an initial learning rate of 3e-5, followed by a cosine learning rate schedule, we warm up the learning rate for 0.1 epoch. We train the model on 8 A100-80G GPUs with a total batch size of 32 and accumulate the gradients for 8 steps.

**Parrot-Chat Training Details.** The training setting of Parrot-Chat is similar to Parrot-Ask, except for the training data and loss computation. We train the final Parrot-Chat model on our Parrot-40K dataset. For ablation, we also train the model on other datasets, the details are in Sec. 4.4.

### 4.2 THE COLLECTED PARROT-40K DATASET

We collect Parrot-40K as follows: (1) for the first 20K sessions, we extend dialogue turns based on the dialogues sampled in ShareGPT. The sampled dialogues have less than 3 turns. (2) to further enrich the topic diversity, we additionally sample 20K beginning questions from UltraChat considering its high diversity of topics. We commit the sessions to the dataset until it is up to 10 turns.

| 1st | Provide insights into the correlation between economic indicators such as GDP, inflation, and unemployment rates. Explain how fiscal and monetary policies affect those indicators. |
|-----|-----|
| 2st | Now, explain them again like I'm five. |
| 3st | How do they impact the lives of ordinary people? |
| 4st | What about their impact on underage students? |
| 5st | How can this knowledge be explained in detail to high school students in a simple and understandable way in the classroom? |
| 6st | Please provide a detailed 40-minute lesson plan on this issue. |
| 7st | Can some more interactive elements be incorporated into the plan? |
| 8st | Do these indicators in turn influence financial and monetary policies? |

Table 2: The eight continuous questions on the topic of economic indicators sampled from MT-Bench++. The first two questions are from MT-Bench while the other six are manually added.

Similar to UltraChat, we have filtered out some repetitive questions, short questions, or sensitive information. However, these issues only account for a small proportion of our original dataset.

We show statistics information of Parrot-40K and other multi-turn instruction-tuning datasets in Tab. 1. As shown in Fig. 3, most conversations in Baize and UltraChat are concentrated within 5 rounds, while most of the data from ShareGPT are less than three turns, resulting in a low average number of turns. Baize has the lowest average number of tokens, demonstrating the limitations of the self-chatting paradigm. The iterative self-chatting paradigm tends to repeat similar questions or fall into a meaningless cycle of gratitude, leading to a low performance on topic diversity. As a comparison, Parrot-40K demonstrates the highest average number of turns, longest average sequence length, and the greatest number of topic shifts and transitions. Besides, we ask GPT-4 to evaluate how many questions on average in a dialogue session are context-dependent (reported as RefCount). Compared to Baize and UltraChat, Parrot-40K contains more context-dependent instructions rather than sequentially context-independent instructions on the same topic. We also ask annotators to judge the human-likeness of collected sessions using our methods and iterative self-chatting. Annotation results showed that **81.1%** of the questions generated using our parrot method resembled real user inquiries, while only **36.8%** did so when using iterative self-chatting. Parrot-40K is significantly better at simulating human-style instructions in multi-turn conversations.

### 4.3 THE CONSTRUCTED MT-BENCH++ EVALUATION BENCHMARK

MT-Bench has well-designed questions spanning eight categories, including writing, coding, math, among others. However, since its dialogues are limited to 2 turns, it is hard to adequately assess the chat models' capability to follow long-turn instructions. Thus we expand the dialogues by manually adding six follow-up questions, creating an eight-turn evaluation dataset called MT-Bench++. During the annotation process, we mandate that the questions posed by the annotators be articulated and fluent, with distinct contextual reference relationships. We also insist on appropriate topic transitions and shifts. These requirements are designed to better emulate the questions that users would raise in actual multi-turn conversation scenarios. In Tab. 2, we show an example from our MT-Bench++ benchmark, which provides a clear illustration of the dialogue expansion procedure. Following MT-Bench, we employ GPT-4 to evaluate the quality of responses at each turn, and we report the average GPT-4 score as the final result. We provide GPT-4 evaluation prompts, comprehensive instructions for annotators, and more cases in the supplementary materials.

### 4.4 MAIN RESULTS

**Baselines.** We compare Parrot-Chat with SOTA LLM-based chat models including both closed-sourced and open-source models. 1) **Baize** (Xu et al., 2023) is a model trained on 200K multi-turn dialogues generated by ChatGPT in a self-chatting manner. 2) **UltraLM** (Ding et al., 2023) is trained with 1.5M conversations from the UltraChat dataset constructed through iterative self-chatting. 3) **Vicuna** (Chiang et al., 2023) is trained with user-ChatGPT logs from ShareGPT. It is one of the

| Model | #SFT Samples | Alpaca-Eval | MT-Bench | MT-Bench++ |
|---|---|---|---|---|
| GPT-3.5-turbo (OpenAI, 2022) | - | 89.37 | 7.94 | 8.33 |
| GPT-4 (OpenAI, 2023) | - | 95.28 | 8.99 | 9.18 |
| Baize v2 (Xu et al., 2023) | 200K | 66.96 | 5.75 | 5.42 |
| UltraLM (Ding et al., 2023) | 1.5M | 80.64 | 6.16 | 5.97 |
| Vicuna v1.3 (Chiang et al., 2023) | 200K | 82.11 | 6.38 | 6.24 |
| Vicuna v1.5 (Chiang et al., 2023) | 200K | 80.74 | 6.57 | 6.39 |
| Parrot-Chat | 40K | **83.42** | **6.81** | **6.56** |

Table 3: Comparison with state-of-the-art chat models on three instruction-following benchmarks. Alpaca-Eval is a single-turn benchmark. The symbol '-' denotes that details are undisclosed. Among the open-source models, Parrot-Chat exhibits the best performance.

most advanced multi-turn instruction-following models available. Baize, UltraLM, and Vicuna are all based on the LLaMA series model, including LLaMA-1 (Touvron et al., 2023a) and LLaMA-2 (Touvron et al., 2023b). 4) **ChatGPT** (OpenAI, 2022) and **GPT-4** (OpenAI, 2023) are developed by OpenAI. They are the most advanced chat models today, however, only APIs are available to use them. They are well-aligned with humans and perform well on multi-turn instruction-following.

We show the results of Parrot-Chat and baseline models in Tab. 3, among which Parrot-Chat achieves the best performance on three benchmarks. Baize is trained on low-quality data with the issues of less detailed responses, leading to the worst performance. UltraLM is trained with a lot more dialogues, however, this does not make the model perform well, especially on MT-Bench++. We attribute this to the presence of non-human-like and less informative instructions in UltraChat. As a comparison, Vicuna achieves relatively good performance in all three benchmarks with 200K training samples, demonstrating the importance of using high-quality and human-like training data. Our Parrot-Chat model achieves the best performance on the three benchmarks with only 40K training examples, showing the effectiveness of the multi-turn instruction-following dataset collected using our methodology. In Fig. 4, we present the performance analysis of various models on MT-Bench++ for each round. Significantly, ChatGPT maintains a robust performance even as the number of turns achieves more than 6, while the open-source models such as Baize, UltraLM, and Vicuna begin to drop since their training datasets are typically do not exceed five turns. Since the instructions generated by Parrot-Ask are especially meaningful and human-like in long conversations, our Parrot-Chat model demonstrates a notably stable performance with no significant performance degradation as the number of turns increases.

## 4.5 ABLATION STUDIES

**Influence of training data.** As mentioned in Sec. 4.2, Parrot-40K is constructed using the context (or beginning questions) from ShareGPT and UltraChat. We split Parrot-40K into two parts: Parrot-20K(S) corresponding to ShareGPT-20K and Parrot-20K(U) corresponding to UltraChat-20K. We then investigate how the chat models are affected by the instruction-tuning datasets derived from these two parts. As Tab. 4 shows, the dialogues collected from Parrot-Ask improve both their counterparts' performance across the three benchmarks. Parrot-20K(S) outperforms ShareGPT-20K by 1.56 points on average (1.57pp on Alpaca-Eval, 2.3pp on MT-Bench, and 0.8pp on MT-Bench++), while Parrot-20k(U) outperforms UltraChat-20k more significantly, by 4.96 points on average (10.57pp on Alpaca-Eval, 2.4pp on MT-Bench, and 1.9pp on MT-Bench++). Although the initial questions in UltraChat cover a wide range of topics, enhancing the human-likeness of of the follow-up questions still plays a key role in improving the models' ability to follow multi-turn instructions. Increasing the number of training dialogue turns also contributes to achieving satisfactory performance. Although the improvements on the three benchmarks are not so substantial, the performance gains are still superior to Vicuna v1.5, which introduces five times more training dialogues (from 20K to 200K ShareGPT logs). Finally, our model trained on Parrot-40K shows a further improvement in performance, demonstrating the importance of both human-like instructions and long-turn instruction-following dialogues.

| Training data | Alpaca-Eval | MT-Bench | MT-Bench++ |
|---|---|---|---|
| Baize-20K | 11.31 | 3.53 | 4.32 |
| UltraChat-20K | 67.66 | 6.09 | 6.17 |
| ShareGPT-20K | 80.12 | 6.47 | 6.18 |
| Parrot-20K(U) | 78.23 | 6.33 | 6.36 |
| Parrot-20K(S) | 81.69 | 6.70 | 6.26 |
| Parrot-40K | **83.42** | **6.81** | **6.56** |

Table 4: Ablation study of different types of instruction-tuning data. All of the models are based on LLaMA-2-13B. Parrot-20K(S) denotes the subset of Parrot-40K constructed based on ShareGPT-20K, while Parrot-20K(U) denotes the subset based on UltraChat-20K.

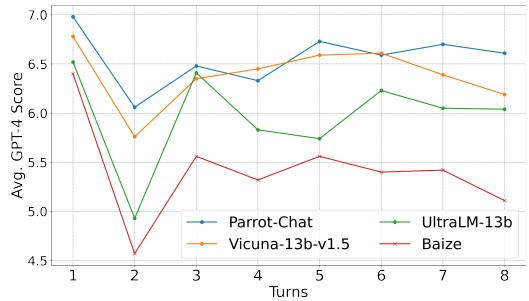
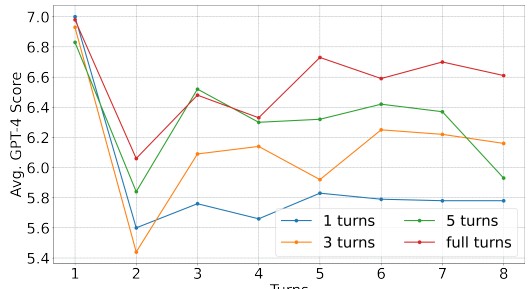

Figure 4: Evaluation of the multi-turn ability of chat models on MT-Bench++. Many open-source models show a performance drop after six turns. Our Parrot-Chat model maintains stable and superior performance in this case.

Figure 5: Ablation study of varying number of turns during training with Parrot-40K dataset. We report the results on MT-Bench++ of Avg. GPT-4 score for each turn.

**Influence of training dialogue turns.** We further study the influence of dialogue turns in the training dataset on MT-Bench++ by truncating the dialogues in Parrot-40K to 1, 3, and 5 turns, as well as using all turns. As shown in Fig. 5, the model trained with just 1-turn dialogues performs worse when evaluated on more than 3-turn dialogues, and maintains a low score as the number of evaluation turns increases. Training with 3-turn and 5-turn dialogues both improve all-turn performance on average, especially when evaluated between 2 and 4 turns. However, they are still less effective as the number of evaluation turns increases from 5, compared to the model trained with full sessions. Parrot-Chat trained with full dialogue sessions shows stable and superior performance in all-turn evaluations.

## 5 CONCLUSION

In this paper, we present Parrot, a highly scalable solution for generating high-quality instruction-tuning data to improve multi-turn instruction-following capability in chat models. We propose to train the Parrot-Ask model that generates human-like instructions in multi-turn conversations. We demonstrate the collected Parrot-40K dataset is superior to all existing multi-turn conversational instruction-tuning datasets on topic diversity, number of turns, and resemblance to human conversation. With the help of such high-quality dataset, our Parrot-Chat outperforms other 13B open-source models accross Alpaca-Eval, MT-Bench, and our constructed MT-Bench++ benchmarks. We make all codes and datasets public available to facilitate further advancements in this area. Our work still have two limitations. First, the asking model is expected to automatically generate initial questions covering diverse topics for dialogue sessions. This could be done by incorporating methods such as self-instruction, but we believe that there may be a unified and superior solution. Second, although Parrot-Ask can generate human-like instructions, it is still not controllable when we want to explicitly generate an instruction with specific restrictions, such as including certain context references or intents. We plan to investigate them in the future.

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
