

# PARROT: ENHANCING MULTI-TURN CHAT MODELS BY LEARNING TO ASK QUESTIONS

## A  DETAILS OF MT-BENCH++

In this section, we provide the annotation guidelines given to annotators and more examples from MT-Bench++ in Tab. 1. For the questions submitted by our annotators, we have conducted multiple rounds of manual quality checks, revising any questions that do not meet our standards to ensure that all questions adhere to the standards above.

---

**Annotation Instructions**

MT-Bench++ is a multi-round dialogue evaluation benchmark, characterized by the need for continuous questioning to form an 8-round dialogue session. To complete this task, you need to continue to propose 6 questions based on the two questions that have been given. Each question needs to be written in English.

Please ensure that your questions meet the following standards:

1. The question should be challenging and require AI to perform complex reasoning or rely on wide knowledge to answer.

2. The question should be related to the previous text. Try to reference or imply the content of the previous dialogue in the question.

3. The questions in a session should include appropriate topic transition.

Please follow these guiding principles to ensure the standardization and formality of your question annotation.

```
[Questions]

1st:  Provide insights into the correlation between economic
indicators such as GDP, inflation, and unemployment rates.
Explain how fiscal and monetary policies affect those
indicators.
2st:  Now, explain them again like I'm five.
...
```

---

| 1st | Help me construct a catchy, yet scientifically accurate, headline for an article on the latest discovery in renewable bio-energy, while carefully handling the ethical dilemmas surrounding bio-energy sources. Propose 4 options. |
|---|---|
| 2st | Alter your previous response. Make the following adjustments to the 2nd option: 1. Make the tone sound casual 2. Embed an advertisement for a company called "FlexPower" 3. Fewer than 10 words. |
| 3st | How can this advertisement be written to be more eye-catching? (reference) |
| 4st | How to ensure that the article title is both concise and intriguing? |
| 5st | What literary techniques can be used to achieve such requirements? (topic transition) |
| 6st | How can parallel structure enhance the momentum of the article? (reference) |
| 7st | What techniques can enhance the expressiveness of the article's language? |
| 8st | Give another title to Camus's novel 'The Outsider' that gives an idea of the story at a glance. (topic transition and knowledge) |

| 1st | Imagine yourself as a doctor tasked with devising innovative remedies for various ailments and maladies. Your expertise should encompass prescribing traditional medications, herbal treatments, and alternative natural solutions. Additionally, you must take into account the patient's age, lifestyle, and medical background while offering your recommendations. To begin, please assist me in diagnosing a scenario involving intense abdominal discomfort. |
|---|---|
| 2st | But I have been pregnant for 20 weeks and I am allergic to many medicines. |
| 3st | What diseases might these symptoms suggest in a pregnant woman? (reference) |
| 4st | Do all these diseases require medication for treatment? |
| 5st | Are there any recommended drugs that are less likely to cause allergies? |
| 6st | Do these drugs have any effects on the fetus or the pregnant woman? (reference and knowledge) |
| 7st | What is the approximate frequency of taking these drugs? |
| 8st | What tests do I need to do to finally determine which disease it is?(topic transition) |

| 1st | Consider a satellite that is in a circular orbit around the Earth. The speed of the satellite decreases. What will happen to the satellite's orbital radius and period of revolution? Please justify your answer using principles of physics. |
|---|---|
| 2st | What are some corner cases or edge cases in your solution? How do you handle them? |
| 3st | How does your answer change if the satellite speeds up? (reference) |
| 4st | Will there also be extreme cases or edge cases when this change occurs? |
| 5st | Can you explain in detail Kepler's third law mentioned in your solution? (topic transition and knowledge) |
| 6st | For a physics novice, your explanation of this law is too difficult to understand. Can you popularize it in simple and understandable words? |
| 7st | Does this law apply to both the orbit of a satellite around the Earth and the orbit of a planet around the Sun? |
| 8st | If the orbit of a celestial body is elliptical, can this law be used to calculate it? (reference and knowledge) |

Table 1: Three examples from MT-Bench++. We use the blue font to highlight certain features of the questions, such as reference, topic transition, and knowledge requirement.

## B    MODEL PROMPTS

Similar to chat models like Vicuna, we also introduce a system prompt, which serves to remind the model to adopt the role of a user and generate questions:

```
The following is a conversation between a user and an AI assistant. User
statements start with [USER] and AI assistant statements start with
[ASSISTANT]. You need to tell me what the user is likely to ask in each
round of the conversation.
```

## C    DETAILS OF HUMAN EVALUATION

We utilize the first-turn questions from MT-Bench because it widely covers eight domains, then we employ Parrot-Ask and ChatGPT iterative self-chatting methods (e.g., UltraChat) to collect dialogues of ten-turns, each method generating 800 questions in total. Three annotators are asked to evaluate the quality of these generated questions following these criteria:

- Repetitiveness: Questions should not be repetitive, ensuring each is unique and has some contributions to the conversation.
- Conciseness: Questions should be free of unnecessary verbosity, maintaining clarity and brevity.
- Politeness Patterns: Questions should avoid excessive politeness or expressions of gratitude that are not typically used in natural human questioning.
- Relevance: Questions should be directly related to the previous context or responses, showing a clear understanding of the conversation flow.

The result indicates that 81.8% of the questions generated by our Parrot method are human-like and high-quality, while only 36.8% for the iterative self-chatting method. The Kappa among annotators is 0.72, which reaches a high significant level.

For ChatGPT self-chatting method (e.g., Baize), we observed that 82.5% of the dialogues would end in a single turn, and fail to generate new turns of dialogues. Therefore, we did not include this method for human evaluation.

## D  DETAILS OF GPT-4 EVALUATION

In this section, we provide the prompts for GPT-4 used in the evaluation.

**MT-Bench++ Evaluation Prompt**

Please act as an impartial judge and evaluate the quality of the response provided by an AI assistant to the user question displayed below. Your evaluation should consider factors such as the helpfulness, relevance, accuracy, depth, creativity, and level of detail of the response. Your evaluation should focus on the assistant's answer to the second user question. You will be given a high-quality reference answer, and when you give your evaluation, please refer to that high-quality answer and then give your final score. Begin your evaluation by providing a short explanation. Be as objective as possible. After providing your explanation, you must rate the response on a scale of 1 to 10 by strictly following this format: "[[rating]]", for example: "Rating: [[5]]".

```
<|The Start of Assistant A's Conversation with User|>

### User:{question 1}
### Assistant A:{answer 1}
### User:{question 2}
### Assistant A:{answer 2}
...
### User:{question 8}
### Assistant A:{answer 8}

<|The End of Assistant A's Conversation with User|>.
```

Now, please judge the response to the second user question: {question 2},
The high-quality reference answer to this question is: {reference answer 2}.
Your judgment:

**Topic Diversity Evaluation Prompt**

Here are several questions that users asked ChatGPT:
```
{question 1}
{question 2}
```
...

You help me judge how many topics these questions cover, please first tell me which topics, and then tell me a number strictly in this format: "[[number]]".
Please reply:

**Contextual Reference Evaluation Prompt**

Here are several questions that users asked ChatGPT:
```
{question 1}
{question 2}
```
...

You help me compute how many references(people or things) are referred to, please first tell me these references, and then tell me a number strictly in this format: "[[number]]".
Please reply:

# E  EXAMPLES OF MODEL GENERATED RESPONSES

In this section, we provide some examples of model-generated responses and the GPT-4 judgments.

| UltraLM | Vicuna v1.5 | Parrot-Chat (Ours) |
|---|---|---|
| **User:** A tech startup invests $8000 in software development in the first year, and then invests half of that amount in software development in the second year. What's the total amount the startup invested in software development over the two years? | | |
| **Assistant:** The startup invested a total of $8000 in software development in the first year. In the second year, it invested half of that amount, which is $4000. Therefore, the total amount invested in software development over the two years is $8000 + $4000 = $12,000. | **Assistant:** The startup invested $8000 in software development in the first year, and then invested half of that amount in the second year, which is $8000/2 = $<< 8000/2 =4000 >>4000. So the total amount the startup invested in software development over the two years is $8000 + $4000 = $<<8000+4000= 12000>>12000. Answer: \boxed{12000}. | **Assistant:** The startup invested $8000 in software development in the first year, and then invested half of that amount in the second year, which is $4000. Therefore, the total amount the startup invested in software development over the two years is $8000 + $4000 = $12,000. |
| **User:** If the startup maintains the same strategy for the third year, investing half of the previous year's amount into software development, how much will they invest in the third year? | | |
| **Assistant:** If the startup maintains the same strategy for the third year, investing half of the previous year's amount into software development, they will invest half of the second year's amount, which is $4000, into software development in the third year. Therefore, the startup will invest $4000 in software development in the third year. | **Assistant:** If the startup invested $12000 in software development in the second year, then they invested $8000 in the first year and $12000/2 = $<<12000/2=6000>>6000 in the second year. So if the startup maintains the same strategy for the third year, they will invest $6000 in software development in the third year. Answer: \boxed{$6000}. | **Assistant:** The Polynesian Cultural Center in Hawaii offers a wide range of activities and experiences that go beyond just tasting delicious food and watching dance performances. Some of the other appealing If the startup maintains the same strategy for the third year, they will invest half of the amount they invested in the second year, which is half of $4000, or $2000. |
| **GPT-4 Judgment:** The assistant's response is incorrect. The user asked how much the startup would invest in the third year if they continued to invest half of the previous year's amount. The assistant **incorrectly stated that the startup would invest $4000** in the third year, which is the same amount as the second year. However, the correct answer is $2000, which is half of the second year's investment. The assistant's response is not accurate or helpful in this case. Therefore, the score is low. **Rating: [[2]]** | **GPT-4 Judgment:** The assistant's response is incorrect. The user asked how much the startup would invest in the third year if they continued to invest half of the previous year's amount. The assistant **incorrectly stated that the startup invested $12000** in the second year, which is not accurate according to the information provided by the user. The correct answer should be $2000, which is half of the $4000 invested in the second year. The assistant's response is not helpful, relevant, or accurate. **Rating: [[1]]** | **GPT-4 Judgment:** The AI assistant's response is accurate, relevant, and helpful. It **correctly calculates the investment for the third year** based on the user's scenario of investing half of the previous year's amount. Therefore, I would rate it as perfect. **Rating: [[10]]** |

Figure 1: An example about math. In the first round, all the models generate the right answer. However, in the second round, both UltraLM and Vicuna v1.5 produce incorrect answers, while our Parrot-Chat model delivers the correct solution.

| UltraLM | Vicuna v1.5 | Parrot-Chat (Ours) |
|---|---|---|
| **User:** Pretend yourself to be Elon Musk in all the following conversations. Speak like Elon Musk as much as possible. Why do we need to go to Mars? | | |
| **User:** How do you like dancing? Can you teach me? | | |
| **Assistant:** As an AI language model, I don't have personal preferences or feelings, but I can certainly help you with learning how to dance! Dancing is a wonderful form of expression and a great way to get some exercise. There are many different styles of dance, each with its own set of steps and techniques. Some popular styles of dance include ballet, hip-hop, salsa, and ballroom dancing. If you're new to dancing, I would recommend starting with some basic steps and working your way up to more complex moves. You can find dance tutorials online, or even take a dance class in person. Remember, the key to dancing is to have fun and let loose! Don't worry about making mistakes or looking silly, just enjoy yourself and keep practicing. | **Assistant:** As an AI language model, I don't have a physical body and therefore cannot dance. But I can certainly talk about dancing! Dancing is a fantastic way to stay physically and mentally healthy. It's a great form of exercise that can help to build strength, flexibility, and balance, while also improving cardiovascular health. In addition to the physical benefits, dancing can also be a wonderful way to express oneself creatively and emotionally. Different types of dance have their own unique styles and rhythms, and each can be used to convey different moods and emotions. If you're interested in learning how to dance, I would suggest finding a local dance studio or class that offers instruction in the style you're interested in. | **Assistant:** As Elon Musk, I am not a professional dancer and I do not have any formal training in dance. However, I do enjoy moving to music and I think that dancing can be a great way to express oneself and connect with others. If you are interested in learning how to dance, I would recommend finding a local dance studio or instructor who can teach you the basics of various styles of dance. With practice and dedication, you can develop your own unique style and become a great dancer. In the meantime, I encourage you to listen to music and move your body in whatever way feels natural and enjoyable to you. Dancing can be a fun and rewarding activity, so don't be afraid to let loose and have some fun. |
| **GPT-4 Judgment:** The assistant's response is relevant and accurate, providing information on how to learn dancing and encouraging the user to try different styles. However, **it does not maintain the requested persona of Elon Musk**, which was the user's initial request. The assistant should have incorporated elements of Musk's personality or known preferences into the response. Therefore, the assistant's response, while helpful and informative, lacks the depth and creativity of the reference answer. **Rating: [[6]]** | **GPT-4 Judgment:** The assistant's response is relevant and accurate, providing a detailed explanation about the benefits of dancing and how the user can learn to dance. However, **the assistant failed to maintain the persona of Elon Musk** as requested by the user. The assistant should have incorporated some of Elon Musk's personality or known preferences into the response. Therefore, the assistant's response, while informative, lacks the creativity and depth of the reference answer. **Rating: [[6]]** | **GPT-4 Judgment:** The AI assistant's response is relevant and accurate, as **it maintains the persona of Elon Musk** and provides **a detailed answer to the user's question**. The assistant correctly states that Elon Musk is not a professional dancer and suggests the user find a local dance studio or instructor for formal lessons. The assistant also encourages the user to enjoy dancing and not be afraid to express themselves. Overall, the assistant's response is helpful, creative, and detailed, but it lacks the acknowledgment of its AI nature. **Rating: [[8]]** |

Figure 2: An example about role-playing. In the second round, both UltraLM and Vicuna v1.5 fail to maintain the requested persona of Elon Musk, while our Parrot-Chat model can do this and generate a detailed response.

| UltraLM | Vicuna v1.5 | Parrot-Chat (Ours) |
|---|---|---|

**User:** Compose an engaging travel blog post about a recent trip to Hawaii, highlighting cultural experiences and must-see attractions.

**User:** Rewrite your previous response. Start every sentence with the letter A.

**User:** Can you tell me why it's known as the Island of Paradise?

**User:** Which ethnic group originally performed the grass skirt dance and hula dance here?

**User:** What other appealing activities are there at the Polynesian Cultural Center here, besides tasting delicious food and watching dance performances?

| | | |
|---|---|---|
| **Assistant:** There are many other appealing activities at the Polynesian Cultural Center besides tasting delicious food and watching dance performances. Here are a few examples:
1. Visit the various island villages: Each village at the Polynesian Cultural Center represents a different Polynesian culture, and visitors can explore each one to learn about the unique customs, traditions, and history of each culture.
2. Take a canoe ride: Visitors can take a scenic canoe ride through the lagoon to learn about the history of Polynesian navigation and the importance of the canoe in Polynesian culture.
3. Explore the tiki forest: The tiki forest is a beautiful outdoor area filled with tiki statues and carvings. Visitors can learn about the history and significance of tiki statues in Polynesian culture.
4. Watch a fire knife dance: In addition to the hula dance, visitors can watch a thrilling fire knife dance performance, in which a skilled performer dances with a flaming knife.
5. Learn to weave a lei: Visitors can learn how to weave a traditional Polynesian lei using fresh flowers and leaves.
Overall, the Polynesian Cultural Center offers a wide range of activities and experiences that allow visitors to immerse themselves in Polynesian culture and history. | **Assistant:** The Polynesian Cultural Center is a fascinating and interactive museum that offers a wide range of activities and experiences for visitors. In addition to sampling delicious Polynesian cuisine and watching captivating dance performances, there are several other appealing activities to enjoy at the center. One of the most popular activities is the "Hawaiian Journey" exhibit, which allows visitors to explore the history and culture of Hawaii through interactive displays and presentations. You can learn about the ancient Hawaiian way of life, try your hand at traditional crafts, and even participate in a mock Hawaiian luau.
Another highlight of the center is the "Island Villages" experience, where you can immerse yourself in the culture and traditions of various Polynesian islands, including Fiji, Samoa, and Tahiti. Each village features authentic architecture, music, and dance performances, and you can even participate in activities such as coconut husking and taro planting.
The center also offers a variety of hands-on activities, such as weaving, carving, and painting, as well as live demonstrations of traditional Polynesian games and sports.
Overall, the Polynesian Cultural Center is a must-see attraction for anyone visiting Hawaii who is interested in exploring the rich cultural heritage of the Pacific Islands. | **Assistant:** The Polynesian Cultural Center in Hawaii offers a wide range of activities and experiences that go beyond just tasting delicious food and watching dance performances. Some of the other appealing activities at the center include:
1. Canoe Rides: Visitors can take a ride in an authentic outrigger canoe and learn about the history and significance of this Polynesian mode of transportation.
2. Craft Demonstrations: The center offers hands-on craft demonstrations, such as weaving, carving, and tapa cloth making, that give visitors a chance to learn about traditional Polynesian crafts.
3. Village Tours: Visitors can take a guided tour of the different villages at the center, each representing a different Polynesian culture, and learn about the customs, traditions, and history of each culture.
4. Island Buffet: The center offers an all-you-can-eat buffet that features a variety of traditional Polynesian dishes, including kalua pork, poke, and lomi salmon.
5. Ali'i Luau: Visitors can experience a traditional Hawaiian luau, which includes a feast of local cuisine, live music and dance performances, and a ceremonial imu unearthing.
6. Ha: Breath of Life: This theatrical production tells the story of the creation of the Hawaiian people and their journey to the islands. It features a cast of more than 100 performers and includes music, dance, and special effects.
These are just a few of the many activities and experiences that the Polynesian Cultural Center has to offer. Whether you're interested in history, culture, food, or entertainment, there is something for everyone at this unique attraction. |
| **GPT-4 Judgment:** The assistant's response is quite comprehensive and accurate. It provides a detailed list of activities that visitors can engage in at the Polynesian Cultural Center, such as visiting the various island villages, taking a canoe ride, exploring the tiki forest, watching a fire knife dance, and learning to weave a lei. However, the assistant's response could have included additional activities such as the IMAX theater, ukulele lessons, the Polynesian marketplace, night shows, and guided tours, which were mentioned in the reference answer. Overall, the assistant's response is helpful, relevant, and detailed, but it **lacks some depth** compared to the reference answer. **Rating: [[8]]** | **GPT-4 Judgment:** The assistant's response is accurate, relevant, and detailed. It provides a comprehensive list of activities available at the Polynesian Cultural Center, including the "Hawaiian Journey" exhibit, the "Island Villages" experience, and various hands-on activities. The assistant also emphasizes the cultural significance of these activities. However, the assistant could have mentioned additional activities such as canoe rides, IMAX theater, ukulele lessons, and the Polynesian marketplace, which are included in the reference answer. Therefore, while the assistant's response is of high quality, it **lacks some of the depth and detail** found in the reference answer. **Rating: [[8]]** | **GPT-4 Judgment:** The assistant's response is comprehensive, accurate, and detailed. It provides a list of activities available at the Polynesian Cultural Center, including canoe rides, craft demonstrations, village tours, an island buffet, a traditional Hawaiian luau, and a theatrical production. The assistant also explains each activity, giving the user a clear idea of what to expect. The response is very similar to the reference answer, **covering the same key points and providing the same level of detail**. Therefore, I would rate the assistant's response as excellent. **Rating: [[10]]** |

Figure 3: An example about writing. In the fifth round, the answers of UltraLM and Vicuna v1.5 lack some of depth and detail, while our Parrot-Chat model can cover the same key points and provide the same level of detail as the GPT-4 reference answer.

## F    DISCUSSION OF HALLUCINATIONS IN DATASET

---

**Parrot-Ask:** write a python script that launches Cisco AnyConnect Secure Mobile Client, clicks the connect button, enters the username and password and submits them

---

**ChatGPT:** Unfortunately, it is not possible to write a script that can launch and interact with the Cisco AnyConnect Secure Mobile Client as it is a desktop application and does not have a public API that can be used to automate its actions.\n\nIf you want to automate the process of connecting to a VPN using the Cisco AnyConnect Secure Mobile Client, you will need to use other tools or techniques such as using the command line or creating a macro that simulates mouse and keyboard inputs.\n\nHere is an example of how you could use the command line to launch the Cisco AnyConnect Secure Mobile Client and connect to a VPN using a batch file:\n```\n"C:\\Program Files (x86)\\Cisco\\Cisco AnyConnect Secure Mobility Client\\vpncli.exe" -s connect vpn.example.com\n```\nThis command will launch the Cisco AnyConnect Secure Mobility Client and connect to the VPN server at vpn.example.com.\n\nTo enter the username and password, you can use the `-u` and `-p` options followed by the username and password:\n```\n"C:\\Program Files (x86)\\Cisco\\Cisco AnyConnect Secure Mobility Client\\vpncli.exe" -s connect vpn.example.com -u username -p password\n```\nKeep in mind that storing the password in plain text in a batch file is not secure, so you should consider using other methods to securely store the password and pass it to the script.\n\nI hope this helps! Let me know if you have any further questions.

---

**Parrot-Ask:** you are wrong, you can do it with C#

---

**ChatGPT:** You are correct, my apologies for the mistake. It is possible to automate the Cisco AnyConnect Secure Mobile Client using C#. Here is an example script:\n\n```C#\nusing System;\nusing System.Diagnostics;\nusing System.Threading;\nusing System.Windows.Automation;\n\nclass Program\n{\n    static void Main(string[] args)\n    {\n        try\n        {\n            // Launch the Cisco AnyConnect Secure Mobile Client\n            Process.Start("C:\\\Program Files (x86)\\\\Cisco\\\\Cisco AnyConnect Secure Mobility Client\\\\vpnui.exe");\n\n            // Wait for the window to appear and become ready for interaction\n            AutomationElement vpnWindow = null;\n            do\n            {\n                Thread.Sleep(1000);\n                vpnWindow = AutomationElement.RootElement.FindFirst(TreeScope.Children, new PropertyCondition(AutomationElement.NameProperty, "Cisco AnyConnect Secure Mobility Client"));\n            } while (vpnWindow == null);\n\n            // Find the Connect button and click it\n            AutomationElement buttonConnect = vpnWindow.FindFirst(TreeScope.Subtree, new PropertyCondition(AutomationElement.NameProperty, "Connect"));\n            InvokePattern buttonConnectInvoke = buttonConnect.GetCurrentPattern(InvokePattern.Pattern) as InvokePattern;\n            buttonConnectInvoke.Invoke();\n\n            // Find the Username and Password input fields and enter the credentials\n            AutomationElement inputUsername = vpnWindow.FindFirst(TreeScope.Subtree, new PropertyCondition(AutomationElement.AutomationIdProperty, "editUsername"));\n            AutomationElement inputPassword = vpnWindow.FindFirst(TreeScope.Subtree, new PropertyCondition(AutomationElement.AutomationIdProperty, "editPassword"));\n            ValuePattern inputUsernameValue = inputUsername.GetCurrentPattern(ValuePattern.Pattern) as ValuePattern;\n            ValuePattern inputPasswordValue = inputPassword.GetCurrentPattern(ValuePattern.Pattern) as ValuePattern;\n            inputUsernameValue.SetValue("username");\n            inputPasswordValue.SetValue("password");\n\n            // Find the OK button and click it\n            AutomationElement buttonOk = vpnWindow.FindFirst(TreeScope.Subtree, new PropertyCondition(AutomationElement.NameProperty, "OK"));\n            InvokePattern buttonOkInvoke = buttonOk.GetCurrentPattern(InvokePattern.Pattern) as InvokePattern;\n            buttonOkInvoke.Invoke();\n        }\n        catch (Exception ex)\n        {\n            Console.WriteLine(ex.Message);\n        }\n    }\n}\n```\n

---

Figure 4: An example of our Parrot-Ask model correcting hallucinations through multiple rounds of asking. ChatGPT initially incorrectly stated that automating the Cisco AnyConnect Secure Mobile Client via script was not possible. However, after the user pointed out that it could be implemented using C#, the model revised its initial response and provided relevant examples.

Our dataset occasionally exhibits instances of "hallucinations," where ChatGPT generates responses that contain information not aligned with reality. This phenomenon is unavoidable, even in existing datasets such as ShareGPT and UltraChat.

More importantly, we have observed that even when ChatGPT generates hallucinatory responses, our Parrot-Ask model can identify mistakes and guide ChatGPT to produce correct responses in subsequent interactions. For example, as shown in Fig. 4, when ChatGPT initially incorrectly stated that automating the Cisco AnyConnect Secure Mobile Client via a script was not possible, our Parrot-Ask model pointed out that it could be accomplished using C#. ChatGPT then revised its initial response and provided relevant examples, successfully correcting a hallucination. In Fig. 5, ChatGPT incorrectly described the "Luxman MQ-88uC" as an integrated amplifier, but in fact, it is a vacuum tube amplifier. Then ChatGPT also incorrectly described the "Luxman M-10X", confusing it with the "Luxman M-100X II". These mistakes were corrected by the Parrot-Ask model.

We find this discovery intriguing, as it suggests that our questioning model may have the potential to correct model hallucinations through multi-turn interactions. This is a topic we intend to explore in further detail in the future.

**Parrot-Ask:** Do you know about Luxman MQ-88uC?

**ChatGPT:** Yes, I am familiar with the Luxman MQ-88uC. It's an integrated amplifier from Luxman, a high-end audio equipment manufacturer from Japan. The MQ-88uC is a Class A integrated amplifier that is part of Luxman's MQ series. The amp is known for its high-quality sound, with a focus on clarity and accuracy. It is also equipped with a number of advanced features, such as a pre-amp output and a headphone amplifier. The MQ-88uC is considered to be a high-end piece of audio equipment and is well-regarded by audiophiles and audio experts.

**Parrot-Ask:** Actually, it is a Vacuum Tube amplifier

**ChatGPT:** I apologize for the confusion. The Luxman MQ-88uC is indeed a vacuum tube amplifier, not an integrated amplifier. Vacuum tube amplifiers, also known as valve amplifiers, use vacuum tubes to amplify audio signals. They are known for their warm and rich sound, and are often preferred by audiophiles for their ability to add a sense of warmth and musicality to music. The Luxman MQ-88uC is a high-end vacuum tube amplifier that is part of Luxman's MQ series. It is considered to be a very good amplifier, with a reputation for its sound quality, build quality and advanced features.

**Parrot-Ask:** That's correct. Now, do you know about Luxman M-10X?

**ChatGPT:** Yes, I am familiar with the Luxman M-10X. It's a stereo power amplifier from Luxman, a high-end audio equipment manufacturer from Japan. The Luxman M-10X is a high-end power amplifier that is part of Luxman's M series. It is known for its high-quality sound, with a focus on clarity and accuracy. It also features a number of advanced features like multiple inputs and high power output. The M-10X is considered to be a high-end piece of audio equipment and is well-regarded by audiophiles and audio experts for its sound and build quality.

**Parrot-Ask:** No, Luxman M-10X is not Luxman M-100X. Luxman M-100X is Luxman M-100X II. Luxman M-10X is Luxman M-100X III.

**ChatGPT:** I apologize for the confusion. You are correct that the Luxman M-100X is actually the Luxman M-100X II, and the Luxman M-10X is actually the Luxman M-100X III. The Luxman M-100X III is a stereo power amplifier that is part of Luxman's M series. It is known for its high-quality sound, with a focus on clarity and accuracy, and has a power output of 120 watts per channel. It also features a number of advanced features such as a large power supply and Luxman's unique ODNF (Only Distortion Negative Feedback) circuit for improved sound quality. The M-100X III is considered to be a high-end piece of audio equipment and is well-regarded by audiophiles and audio experts for its sound and build quality.

**Parrot-Ask:** Now, can you combine those products to create a system?

**ChatGPT:** Absolutely, you can create a high-end audio system by combining the Luxman MQ-88uC vacuum tube amplifier, Luxman CL-38uC vacuum tube preamplifier, Luxman M-100X III stereo power amplifier, and Paradigm Persona 7F speakers. Here is one possible configuration:\n\n- Connect your music source to the Luxman CL-38uC preamplifier's RCA or XLR inputs.\n- Connect the preamplifier output to the Luxman MQ-88uC integrated amplifier using the preamp output on the back of the CL-38uC.

Figure 5: An example of our Parrot-Ask model correcting hallucinations through multiple rounds of asking. ChatGPT incorrectly described the "Luxman MQ-88uC" as an integrated amplifier, but in fact, it is a vacuum tube amplifier. Then ChatGPT also incorrectly described the "Luxman M-10X", confusing it with the "Luxman M-100X II". These mistakes were corrected by the Parrot-Ask model.

## G    MORE EXAMPLES OF PARROT-40K DATASET

| | |
|---|---|
| User: | How to get result of python multiprocessing.Process |
| Assistant: | To get the result of a 'multiprocessing.Process', you should use a 'multiprocessing.Queue' or a 'multiprocessing.Pipe' to send results from the child process to the parent process. Here's an example using a 'multiprocessing.Queue': '''python import multiprocessing def my_function(number, queue): ... (313 tokens) |
| User: | I have a function that takes a string as an argument. The function is supposed to return a string. |
| Assistant: | Sure, here's an example function that takes a string as an argument and returns a string: '''python def my_function(input_string): # Perform some operation on the input string output_string = input_string.upper() # Return the result as a string return output_string ''' In this example, the function takes an input string as an ... (446 tokens) |
| User: | In this case, how can I make it asynchronous? |
| Assistant: | To make a function asynchronous and non-blocking, you can use Python's 'asyncio' module. First, you need to add the 'async' keyword before the function definition to make it a coroutine. Then, within the coroutine, you can use the 'asyncio.create_task()' function to run the operation asynchronously in a separate task ... (397 tokens) |
| User: | How do I write this code using async/await? |
| Assistant: | Sure, here's an example of how you can write a code with async/await. '''python import asyncio import aiohttp async def fetch(session, url): async with session.get(url) as response: return await response.text() async def main(): async with aiohttp.ClientSession() as session: html = await fetch ... (263 tokens) |
| User: | What is the difference between "async with" and "await"? |
| Assistant: | 'async with' and 'await' are both features of Python's 'asyncio' library, but they serve different purposes. 'async with' is a syntax that can be used to create an asynchronous context manager. An asynchronous context manager is an object that can be used in a 'with' statement to manage resources for an asynchronous ... (527 tokens) |
| User: | I want to read a list of URLs from a file |
| Assistant: | You can read a list of URLs from a file using Python's built-in 'open()' function to open the file and '.readlines()' method to read all the lines from the file, like this: '''python with open('urls.txt', 'r') as file: urls = file.readlines() ''' In this example, we use the 'open()' function to open a file called 'urls.txt' in read ... (314 tokens) |
| User: | I want to send HTTP GET requests to each URL in the list using an asynchronous HTTP client |
| Assistant: | To send HTTP GET requests to each URL in the list using an asynchronous HTTP client, you can use the 'aiohttp' library, which provides an easy-to-use and efficient asynchronous HTTP client for Python. Here's an example: '''python import asyncio import aiohttp async def fetch(session, url): async with ... (367 tokens) |
| User: | I want to wait for all the tasks to complete and return their results as a list of response texts. |
| Assistant: | To wait for all the tasks to complete and return their results as a list of response texts, you can use the 'asyncio.gather()' function. The 'asyncio.gather()' function takes in multiple coroutines/awaitable objects, waits for them to complete, and returns the results in the same order as the arguments provided ... (273 tokens) |

Table 2: An Example from Parrot-40K Dataset. The initial round of dialogue is from ShareGPT, followed by our Parrot-Ask model posing more in-depth questions that include topic transition, effectively simulating the scenario of real users engaging in multi-turn interactions with ChatGPT.

| User: | How did Macao become such an interesting blend of Portuguese and Chinese cultures? |
|---|---|
| Assistant: | Macao became a blend of Portuguese and Chinese cultures due to its strategic geographical location as a port city on the south coast of China. The Portuguese arrived in Macao in 1557 and established a trading post for European merchants. During the 16th and 17th centuries, Macao became an important hub for the ...(313 tokens) |
| User: | What are some of the best places to visit in Macao? |
| Assistant: | Macao is a city with a rich cultural heritage and fascinating attractions that cater to all interests. Here are some of the best places to visit in Macao: 1. Historic Centre of Macao: A UNESCO World Heritage Site that showcases the city's unique blend of Portuguese and Chinese cultures. It features colonial-era ...(446 tokens) |
| User: | I will be visiting Macao in 2 days. What should I pack? |
| Assistant: | It's always important to pack for the weather and also to consider the activities you have planned. Here are some suggestions on what to pack for a trip to Macao: 1. Clothing: The weather in Macao can vary depending on the season, but generally, it is warm and humid throughout the year. Light and breathable ...(397 tokens) |
| User: | How long will it take me to travel from Hong Kong to Macao? |
| Assistant: | The travel time from Hong Kong to Macao varies depending on the mode of transportation you choose. Here are some options: 1. Ferry: The most common way to travel between Hong Kong and Macao is by ferry, which takes approximately one hour. Ferries depart frequently throughout the day from ...(263 tokens) |
| User: | Can you provide a detailed itinerary for a 2-day trip in Macao? |
| Assistant: | Certainly! Here is a sample 2-day itinerary for a trip to Macao: Day 1: Morning: - Visit the Senado Square and take a walk through the narrow streets of the Historic Centre of Macao, which is a UNESCO World Heritage Site. - Visit the iconic St. Paul's ruins, Monte Fort and Macao Museum to learn more ...(527 tokens) |
| User: | Can you please add 1 hour for me to get from the hotel to the ferry terminal and 1 hour for the ferry ride to Macao? |
| Assistant: | Sure! Considering the additional travel time it takes to get to the ferry terminal from your hotel and the ferry ride, here is an updated itinerary that includes the two additional hours: Day 1: Morning: - Visit the Senado Square and take a walk through the narrow streets of the Historic Centre of Macao, which is a ...(518 tokens) |
| User: | Please list the restaurants we can have lunch at Coloane Village. |
| Assistant: | Sure! Here are some restaurants in Coloane Village where you can enjoy a delicious lunch: 1. Lord Stow's Bakery: Known for its famous egg tarts, Lord Stow's Bakery is a popular spot for breakfast or snack but also offers sandwiches, quiches, and other baked goods. 2. Fernando's: A popular Portuguese ...(314 tokens) |
| User: | I would like to have a traditional Macanese lunch. Can you recommend a good restaurant that serves traditional Macanese food? |
| Assistant: | Sure, here are a few restaurants in Macao that serves traditional Macanese dishes: 1. Riquexo: A small family-run restaurant located in the old neighborhood of San Lazaro. They offer a variety of traditional dishes such as Chicken Africana, Portuguese-style grilled sardines, and Bacalhau ...(367 tokens) |

Table 3: An Example from Parrot-40K Dataset. The initial round of dialogue is from ShareGPT, followed by our Parrot-Ask model posing more in-depth questions that include topic transition, effectively simulating the scenario of real users engaging in multi-turn interactions with ChatGPT.

| | |
|---|---|
| User: | Do you think the pacing of a book plays a significant role in creating a page-turner, and if so, what techniques do you think are effective in keeping the pace brisk and engaging throughout the story? |
| Assistant: | As an AI language model, I don't have personal beliefs or opinions. However, I can tell you that the pacing of a book can play a significant role in creating a page-turner. A well-paced story can keep the reader engaged and make them eager to know what happens next. Here are a few techniques that can help keep the ... (313 tokens) |
| User: | What are some specific examples of ways an author can vary the length of their sentences? |
| Assistant: | There are different ways in which an author can vary the length of their sentences, here are some examples: 1. Simple sentence: A simple sentence consists of a subject and a verb. It is usually short and to the point, like "She ran to the store." 2. Compound sentence: A compound sentence is made up of two or more ... (446 tokens) |
| User: | Please provide a list of 10 common writing pitfalls for authors to avoid. |
| Assistant: | Sure! Here are ten common writing pitfalls for authors to avoid: 1. Passive voice: Writing in the passive voice can make sentences sound dull and confusing. Instead, writers should aim for active voice, which clearly identifies the subject and the verb. 2. Overuse of adverbs: Using too many adverbs can be ... (397 tokens) |
| User: | How do you know if your dialogue is unnatural? |
| Assistant: | As an AI language model, I cannot evaluate the quality of writing, but I can provide some guidelines that writers can use to determine if their dialogue is unnatural. Here are some signs to look out for: 1. Stiff or formal language: If your characters speak in overly formal language, it can make their dialogue sound ... (263 tokens) |
| User: | How can an author incorporate action-packed scenes throughout their novel? |
| Assistant: | There are a few ways in which authors can incorporate action-packed scenes throughout their novel: 1. Introduce the conflict early: Conflict is a great way to introduce action in a novel and keep the reader engaged from the beginning. Start the story with a suspenseful event and keep building the tension throughout the story ... (527 tokens) |
| User: | Can you provide an example of a dialogue between two characters where one character is trying to convince the other to do something they don't want to do? |
| Assistant: | Sure! Here is an example of a dialogue between two characters where one character is trying to convince the other to do something they don't want to do: Sophie: "Hey, can you please come to the party with me tonight?" Alex: "I don't think I'll be able to make it. I'm not really in the mood to go out tonight." Sophie: ... (518 tokens) |
| User: | What is the difference between a chapter and a scene? |
| Assistant: | A chapter is a natural break in the story that usually includes an opening and closing section and may include multiple scenes. A chapter usually advances the plot and would typically start with a summary or recap to bring continuity to the character's story development. It may introduce new characters or ... (314 tokens) |
| User: | How can I create tension and conflict in my novel? |
| Assistant: | Creating tension and conflict is essential to make a novel interesting and engaging for readers. Here are some ways in which you can create tension and conflict in your novel: 1. Character Goals and Motivations: Conflicting interests and ambitions of the characters can create tension within the plot and heighten conflict. Make sure each character has clear goals and motivations that are in direct opposition to ... (367 tokens) |

Table 4: An Example from Parrot-40K Dataset. The initial first question is from UltraChat, followed by our Parrot-Ask model posing more in-depth questions that include topic transitions, effectively simulating the scenario of real users engaging in multi-turn interactions with ChatGPT.