# OpenReview forum: "Parrot: Enhancing Multi-Turn Chat Models by Learning to Ask Questions"
_ICLR.cc/2024/Conference — Submitted to ICLR 2024_

### Official Review · Reviewer_myqY · 2023-10-27

**Soundness:** 3 good
**Presentation:** 3 good
**Contribution:** 2 fair
**Rating:** 3
**Confidence:** 5

**Summary:**

The paper explores the multi-turn instruction-following capabilities of chat models. The authors propose a method called Parrot-Ask to generate high-quality instruction-tuning data with more human-style instructions. They also introduce a multi-turn evaluation benchmark called MT-Bench++ to assess the performance of chat models. The experiments show that the Parrot-Chat model, trained on the Parrot-Ask data, outperforms other open-source models on various instruction-following benchmarks. The main contributions of the paper include the systematic exploration of multi-turn instruction-following, the development of the Parrot-Ask method, the construction of the MT-Bench++ evaluation benchmark, and the demonstration of the effectiveness of the proposed approach.

**Strengths:**

+ The paper is well-written and clear to read.

+ The proposed method achieves better performance than several strong baselines.

+ The paper identifies the shortage of previous methods (self-chatting and iterative self-chatting) for SFT data generation.

**Weaknesses:**

- For the first contribution the authors claimed, i.e., "show that a high-quality instruction-tuning dataset plays a key role in empowering the multi-turn instruction-following capabilities of the chat models", I think this is obvious enough and has been revealed by many previous works. Personally, I don't take it as a "contribution".

- The paper identifies an important shortage of previous methods (self-chatting and iterative self-chatting), but the proposed method lacks intrinsic novelty. I like the idea that we probably need a better model to simulate real human questions; this is interesting, though, but more like an engineering trick, not a scientific research problem.

- It's very strange to me that the authors "extend dialogue turns based on the dialogues sampled in ShareGPT". I think most of the dialogues in ShareGPT are already finished and it is unnatural to "extend" such dialogues. Extending such dialogues on purpose could make the generated data longer, but not real.

- The generated dataset heavily relies on existing two datasets (ShareGPT and UltraChat), the success of Parrot may be largely owing to the high quality of existing datasets.

- It happens that the performance improvement is because the GPT evaluator prefers the generation that has long content and has multiple rounds of dialogue. I mean the performance is better probably because the model is biased towards generating longer answers, not better answers. Some experiments are needed to further verify this point.

- Minor: given the rapid development of SFT for open-source LLMs, the current SOTA has been leveled to 95%+, even higher than GPT-4. However, the paper only compares relatively weak baselines, I think the author could add several recent baselines to further demonstrate the quality of the proposed dataset.

**Questions:**

Do you have plans to test on larger version of LLaMA?

---

> ### Author Response · Authors · 2023-11-20
> **Response To Reviewer myqY Part 1**
>
> We sincerely thank the reviewer for the valuable comments. We hope our rebuttal will address the concerns of the reviewer.
>
> ### **Q1: Contributions.**
> Although intuitive, previous works have not provided quantitative evidence to support or verify that chat models' ability to follow multi-turn instructions needs high-quality multi-turn data. Existing evaluation benchmarks like Alpaca Eval only cover single-turn instructions, and MT-Bench is limited to two turns. To the best of our knowledge, we are the first to quantitatively assess the multi-turn capabilities of current chat models and to reveal the gaps between them and GPT-4, ChatGPT. We think this should be considered a contribution.
>
> ### **Q2: Values of asking model.**
> The evolution of asking models and chat models should progress in parallel, akin to the interplay between the spear and the shield. An asking model that can provide diverse, in-depth, and extensive questions is indeed critical for effectively evaluating chat models, as it challenges them to prove their ability to handle a wide range of topics and ensures that issues of safety are thoroughly tested. A strong asking model can drive the progress of chat models by expanding their capability boundaries and revealing potential areas for improvement. Therefore, it is imperative for the academic community to explore building more powerful asking models that can drive the progression of chat models towards higher levels of reliability, safety, and versatility.
>
> In this paper, we propose training an asking model and exploring its application in enhancing the multi-turn capabilities of chat models. Our experimental results reveal that an asking model is capable of generating human-like questions, which can help improve a chat model's multi-turn abilities. Furthermore, we believe that training better asking models will have even more applications in the future.
>
>
> ### **Q3: Using ShareGPT.**
> We would like to clarify that the purpose of extending dialogues from ShareGPT is not merely to increase their length, but to enhance their depth and topic diversity. Dialogues in real life do not have predetermined endpoints. In real-life communications, it is common for users to not obtain the information or answers they need in just one round of conversation. They may delve deeper into a topic or transition to related new topics.
>
> Our method, therefore, employs an asking model specifically trained to generate contextually appropriate and topically relevant questions, thereby ensuring a natural progression of the conversation. When provided with short dialogues from ShareGPT, our asking model is capable of introducing new questions that deepen the existing topic or smoothly transition to a new related one. This is in contrast to the approach of simply concatenating short dialogue sessions, which often results in incoherence and logical inconsistencies [R1]. Our data collection method, therefore, not only increases the length of the dialogues but also improves their depth, and the naturalness of topic transitions.
>
> Furthermore, the quality of the data collected through our method is tailored to meet the complex requirements of users in multi-turn dialogue scenarios. The resulting data is not just longer; it is richer and more representative of genuine human conversation patterns, enabling the trained model to better understand and respond to users' needs. To vividly demonstrate the effectiveness of our asking model, we have included additional examples from our dataset in the appendix (**Section G**).
>
> [R1] Re3Dial: Retrieve, Reorganize and Rescale Conversations for Long-Turn Open-Domain Dialogue Pre-training, EMNLP 2023.

---

> > ### Author Response · Authors · 2023-11-20
> > **Response To Reviewer myqY Part 2**
> >
> > ### **Q4: Relying on existing datasets.**
> > Our achievement should not be overly attributed to the reliance on existing datasets such as ShareGPT and UltraChat. We incorporated a portion of short dialogues from ShareGPT and initial questions from UltraChat primarily for the purpose of fair comparison and controlled variables during scientific experiments. In practical applications, we can leverage prompts obtained from real users or self-instruct generated instructions as Alpaca, regardless of whether they have subsequent conversations. This flexibility allows our proposed method to generate larger-scale multi-turn data. Furthermore, the experimental results clearly demonstrate that models trained on the Parrot-40K dataset constructed using our approach outperform those trained on ShareGPT-200K and UltraChat-1.5M datasets, thus validating the effectiveness of our methodology.
> >
> >
> > ### **Q5: Longer answer bias.**
> > Our datset has longer examples because of the increased number of turns, not the length of answer in each turn. The length of our model's answer for each question is not significantly different from that of Vicuna. In Table R2, we show detailed the performance of each model, along with the average length of the answers indicated in parentheses. This indicates that our model indeed achieves higher scores due to the answers of better quality.
> >
> > Table R2. Comparison with state-of-the-art chat models with answer lengths indicated in parentheses. The answer length is measured in characters following Alpaca-Eval.
> > | Model           | Alpaca-Eval  | MT-Bench    | MT-Bench++  |
> > |-----------------|:------------:|:-----------:|:-----------:|
> > | GPT-3.5-Turbo   | 89.37 (827)  | 7.94 (893)  | 8.33 (1458) |
> > | GPT-4           | 95.28 (1365) | 8.99 (1147) | 9.18 (1081) |
> > | Baize-v2-13B    | 66.96 (930)  | 5.75 (994)  | 5.42 (1077) |
> > | UltraLM-13B     | 80.64 (1087) | 6.16 (895)  | 5.97 (979)  |
> > | Vicuna-v1.3-13B | 82.11 (1132) | 6.38 (1066) | 6.24 (1064) |
> > | Vicuna-v1.5-13B | 80.74(1066)  | 6.57 (1030) | 6.39 (1103) |
> > | Parrot-Chat-13B (ours) | 83.42 (1082) | 6.81 (1063) | 6.56 (1157) |
> > | Parrot-Chat-70B (ours) | 89.04 (1044) | 7.78 (946)  | 7.45 (1087) |
> >
> > ### **Q6: More baselines.**
> > We have compared our work to equitable baselines, such as those without the use of larger models, not optimized with RLHF, open-sourced for reproducibility or with an available API.
> > For the leading models on the Alpaca-Eval leaderboard, RLHF is usually applied, and their responses are significantly longer. In Table R3, we have listed some of the model performances and details from the Alpaca Eval leaderboard.
> > Without the help of RLHF, we fine-tune a Parrot-Chat model with 70B parameters, called Parrot-Chat-70B, solely with SFT on our Parrot dataset. Our Parrot-Chat-70B model substantially improves the performance over our Parrot-Chat-13b model from 83.42 to 89.04, which is near the level of GPT-3.5-Turbo (i.e., ChatGPT). This further validates the strength of our dataset.
> >
> > Table R3. Comparison with more baselines on Alpaca Eval.
> > | Model               | Alpaca-Eval | Answer Length | w/ RLHF |
> > |---------------------|:-----------:|:-------------:|:-------:|
> > | GPT-4 Turbo         |    97.70    |      2049     |   yes   |
> > | XwinLM 70b          |    95.57    |      1775     |   yes   |
> > | GPT-4               |    95.28    |      1365     |   yes   |
> > | LLaMA2 Chat 70B     |    92.66    |      1790     |   yes   |
> > | GPT-3.5-Turbo       |    89.37    |      827      |   yes   |
> > | Parrot-Chat-70B (ours)|  89.04    |      1044     |    no   |
> > | Humpback LLaMa2 70B |    87.94    |      1894     |    no   |
> > | Parrot-Chat-13B (ours)|  83.42    |      1082     |    no   |
> >
> > ### **Q1.7 Larger version of LLaMa.**
> > We fine-tune the LLaMA-2-70B model using our Parrot-40K and achieve better performance, with the results presented in **Table R2** and **Table R3**.

---

> > > ### Author Response · Authors · 2023-11-23
> > >
> > > Dear Reviewer myqY,
> > >
> > > Thank you for your valuable feedback during the review process. As the ICLR rebuttal deadline nears, we kindly request your feedback on our responses to ensure we have addressed all concerns. We are ready to provide further clarifications if needed.
> > >
> > > Thanks,
> > > The Authors

---

### Official Review · Reviewer_sPEb · 2023-10-30

**Soundness:** 2 fair
**Presentation:** 3 good
**Contribution:** 2 fair
**Rating:** 5
**Confidence:** 4

**Summary:**

This paper highlights a gap in multi-turn conversation quality between open-source chat models and state-of-the-art closed source (e.g.,  ChatGPT), attributing it to the lack of high-quality instruction-tuning data. For instance most of existing open-source models are trained with single turn dialogues rather than complex multi-turn or topic switching examples. To address this, the authors introduce "Parrot," a that generates high-quality instruction-tuning data from ChatGPT, leading to the development of "Parrot-Chat," a model that significantly improves multi-turn conversation performance.

**Strengths:**

Quality/clarity
- the paper is overall well written and clear. The figures and tables are easy to follow, and the main methodology is clearly explained.
- the proposed models outperform existing baselines of the same (or higher) parameter size.

Significance
- building an high quality multi-turn conversational datasets is definitely very important for building high-quality models.

**Weaknesses:**

Originality
- using ChatGPT generated to train/distill another model has been already widely explored by many other papers. Moreover, it is worth pointing out that using ChatGPT generated dataset has little or no values at this points because: 1) cannot be used for any commercial models, and 2) doesn't unveil how to actually collect high quality datasets.

**Questions:**

check weakness.

**Details Of Ethics Concerns:**

The paper released data from ChatGPT, which might break "usage and terms" if not properly licensed.

---

> ### Author Response · Authors · 2023-11-20
> **Response To Reviewer sPEb**
>
> We sincerely thank the reviewer for the valuable comments. We hope our rebuttal will address the concerns of the reviewer.
>
> ### **Q1: Novelty.**
> Our work is different from previous works that train models with ChatGPT-generated data. We clarify our novelty and significance to the community in the following aspects:
> 1. we analyze the issues of existing multi-turn datasets such as non-human-like instructions, less detailed responses, short sessions, or limited topic transitions.
> 2. to overcome these issues, we propose to train an asking model that can simulate user questions and then collect a new multi-turn dataset. The experimental results demonstrate that our dataset surpasses existing datasets in terms of topic diversity, number of turns, and human-likeness.
> 3. we introduce the new MT-Bench++ benchmark, designed to quantitatively evaluate a model's multi-turn performance. By utilizing our Parrot dataset with more dialogue turns and higher quality, we have enhanced the multi-turn capability of a chat model based on LLaMA-2.
>
>
> ### **Q2: Values of our work.**
> As academic researchers, our primary focus is know-how, e.g., how to mitigate the gap between open source models and the state-of-the-art API models (e.g., GPT-4 and ChatGPT) in multi-turn capability. For this, we indicate that existing multi-turn datasets have some issues and propose to train an asking model to help collecting a multi-turn dataset with superior quality. Our model trained with our dataset show significant improvement on multi-turn capability compared with models trained on ShareGPT, Baize or UltraChat. Our dataset is fully available for faciliting future academic research, akin to UltraChat and Baize.
>
> Moreover, we believe our proposed method of training a human-like asking model can be applied to generate multi-turn data if using an alternative model that permit commercial usage when necessary. For companies, even in cases where manual annotation is involved in collecting multi-turn data, our asking model can be utilized to enrich topic diversity and prolong the duration of conversation. This ensures that annotators refrain from prematurely ending dialogues or encountering limited topic transitions.
>
>
> ### **Q3: Revealing how to collect high-quality dataset.**
> We have delved deeper into the issues present in current multi-turn datasets, such as non-human instructions, less detailed responses, short sessions, or limited topic transitions. Our approach effectively overcomes these issues by employing a specially trained 'asking model' that can promote richer dialogues. This model is designed to simulate user asking, which in turn prompts more human-like questions, longer sessions and broader topics. Through the improved performance of the chat model, manual evaluation and qualitative examples, it is proved that our method can collect higher-quality multi-turn data than existing methods.

---

> > ### Author Response · Authors · 2023-11-23
> >
> > Dear Reviewer sPEb,
> >
> > Thank you for your valuable feedback during the review process. As the ICLR rebuttal deadline nears, we kindly request your feedback on our responses to ensure we have addressed all concerns. We are ready to provide further clarifications if needed.
> >
> > Thanks, The Authors

---

### Official Review · Reviewer_NNDS · 2023-10-31

**Soundness:** 3 good
**Presentation:** 3 good
**Contribution:** 2 fair
**Rating:** 6
**Confidence:** 4

**Summary:**

In this paper, the authors propose a solution to generate instruction-tuning data for multi-turn chat. They first train the Parrot-Ask model to generate questions, conditioning on answers and conversational history. Then, they employ Parrot-Ask to interact with GPT-3.5, collecting multi-turn instruction tuning data. The authors utilize the collected Parrot-40K dataset to train a chat model called Parrot-Chat, which outperforms existing datasets in terms of statistics and performs better on instruction-following benchmarks, including MT-Bench++, an extended version of MT-Bench.

**Strengths:**

- A simple and effective method is proposed to collect multi-turn instruction-tuning data.
- The collected Parrot-40k datasets show larger average number of turns, token length, topic shifts and transitions than other datasets.
- A new benchmark MT-Bench++ is proposed which is an expansion of MT-Bench where additional six follow-up questions are added.
- Experimental results show that Parrot-Chat achieves the best performance on multiple instruction-following benchmarks over open-source models.

**Weaknesses:**

- The human evaluation part is unclear.
- The authors do not reveal the structure of the proposed prompts.
- The authors do not explain how does  the follow-up questions in MT-Bench++ are decided.

**Questions:**

- It seems that the supplementary materials mentioned at the end of section 4.3 are missing.
- For human evaluation, how is the criteria defined? And how many annotators are involved? And how about the sample size and inter-agreement?

---

> ### Author Response · Authors · 2023-11-20
> **Response To Reviewer NNDS**
>
> We sincerely thank the reviewer for the valuable comments.  We are encouraged by the recognition given to our work. We hope our rebuttal will address the concerns of the reviewer.
>
> ### **Q1: Details of human evaluation.**
> We utilize the first-turn questions from MT-Bench because it widely covers eight domains, then we employ Parrot-Ask and ChatGPT iterative self-chatting methods (e.g., UltraChat) to collect dialogues of ten-turns, each method generating 800 questions in total. Three annotators are asked to evaluate the quality of these generated questions following these criteria:
> - Repetitiveness: Questions should not be repetitive, ensuring each is unique and has some contributions to the conversation.
> - Conciseness: Questions should be free of unnecessary verbosity, maintaining clarity and brevity.
> - Politeness Patterns: Questions should avoid excessive politeness or expressions of gratitude that are not typically used in natural human questioning.
> - Relevance: Questions should be directly related to the previous context or responses, showing a clear understanding of the conversation flow.
>
> The result indicates that 81.8% of the questions generated by our Parrot method are human-like and high-quality, while only 36.8% for the iterative self-chatting method. The Kappa among annotators is 0.72, which reaches a high significant level.
>
> For ChatGPT self-chatting method (e.g., Baize), we observed that 82.5% of the dialogues would end in a single turn, and fail to generate new turns of dialogues. Therefore, we did not include this method for human evaluation.
> We will add these important details in the final version of our paper.
>
>
> ### **Q2: Proposed prompts.**
> We use the following prompts for the Parrot-Ask model: "The following is a conversation between a user and an AI assistant. User statements start with [USER] and AI assistant statements start with [ASSISTANT]. You need to tell me what the user is likely to ask in each round of the conversation."
> We have updated an appendix to include this important information (**Section B**). Please refer to the supplementary material.
>
> ### **Q3: Follow-up questions in MT-Bench++.**
> We decide the follow-up questions in MT-Bench++ according to the following standards:
> 1. The question should be challenging and require AI to perform complex reasoning or rely on wide knowledge to answer.
> 2. The question should be related to the previous text. Try to reference or imply the content of the previous dialogue in the question.
> 3. The questions in a session should include appropriate topic transitions.
>
> We have updated an appendix to include this important information and provided more examples of MT-Bench++ (**Section A**), where we have underscored specific features of the questions such as references, topic transitions, and knowledge requirements. These features are highlighted in blue and accompanied by annotations for clarity. For further details, please refer to the supplementary material.
>
>
> ### **Q4: Missing supplementary materials.**
> We apologize for missing the supplementary materials. We have updated an appendix, which introduces more details of MT-Bench++ and evaluations, and more qualitative examples. Please refer to the supplementary material.

---

> > ### Comment · Reviewer_NNDS · 2023-11-22
> >
> > Thanks for the authors' response! The information provided in the appendix is certainly valuable for understanding the paper. Taking into account the feedback from other reviewers, I will maintain the current score.

---

> > > ### Author Response · Authors · 2023-11-23
> > > **Response To Official Comment by Reviewer NNDS**
> > >
> > > We are very grateful for the constructive comments you have provided during the review process. Thank you for recognizing our work. We will revise the paper in accordance with the content of the rebuttal. Once again, we sincerely appreciate the effort you have expended.

---

### Official Review · Reviewer_tuRa · 2023-11-01

**Soundness:** 3 good
**Presentation:** 3 good
**Contribution:** 3 good
**Rating:** 8
**Confidence:** 4

**Summary:**

This paper proposes Parrot, a model trained specifically for simulating a user, on ShareGPT data. The chat model trained with Parrot, Parrot-Chat, outperforms models trained with ChatGPT self-chat data and models trained with ShareGPT data alone.

**Strengths:**

1. This paper is overall well-written and very clear.
2. Different from Baize/UltraChat, which asks ChatGPT to act like a user in a zero-shot manner, the authors trained a user-simulating model with real user prompts. This model serves as a data augmentation tool, especially for very long dialogue.
3. In contrast to ShareGPT, Baize and UltraChat data, the data generated by Parrot can be very long, which allows long-context alignment. This is promising as the model will suffer less out-of-distribution problems for long-context model, e.g., GPT-4 64k, Claude 100k etc. I recommend the authors to emphasize this strength in their paper.

**Weaknesses:**

1. The technical novelty may be limited.
2. I'd like to see experiments with long-context models, e.g., Long LLaMA.

**Questions:**

1. How good is Parrot's out-of-domain performance? For example, how good is Parrot for specific domains? Also I'd like to see more examples. Consider adding an appendix for qualitative examples.
2. I would like to see discussion/analysis for the hallucinations in the data.

---

> ### Author Response · Authors · 2023-11-20
> **Response To Reviewer tuRa Part 1**
>
> We sincerely thank the reviewer for the valuable comments.  We are encouraged by the recognition given to our work. We hope our rebuttal will address the concerns of the reviewer.
>
> ### **Q1: Novelty.**
> We clarify our novelty and significance to the community in the following aspects:
> 1. we analyze the issues of existing multi-turn datasets such as non-human-like instructions, less detailed responses, short sessions, or limited topic transitions.
> 2. to overcome these issues, we propose to train an asking model that can simulate user questions and then collect a new multi-turn dataset. The experimental results demonstrate that our dataset surpasses existing datasets in terms of topic diversity, number of turns, and human-likeness.
> 3. we introduce the new MT-Bench++ benchmark, designed to quantitatively evaluate a model's multi-turn performance. By utilizing our Parrot dataset with more dialogue turns and higher quality, we have enhanced the multi-turn capability of a chat model based on LLaMA-2.
>
>
> ### **Q2: Long-context models.**
> We fully recognize that long-context models require long conversations for alignment. However, due to limitations of ChatGPT, such as the 16K token support, as well as the high costs associated with collecting very long dialogues, as an initial research exploration, we have constructed dialogues of up to ten turns, most of which are around 4K in length. We have demonstrated the effectiveness of our methodology for long conversations through sufficient experiments. Given the emergence of more powerful models like GPT-4-Turbo-128K, we believe our methods will be applicable for the alignment of long-context models, and we plan to explore this in the future.
>
>
> ### **Q3: Out-of-domain performance.**
> MT-Bench has questions across 8 domains. In Table R1, we compare our proposed Parrot-Chat model with OpenAI GPTs and several open-source models across these domains. Although there remains a significant gap compared to GPT-4 overall and across individual domains, our Parrot-Chat-13B model outperforms GPT-3.5-Turbo in the roleplay and humanities domains. Our model also performs the best among all open-source models in 6 out of 8 domains and the second best in the other two domains. In addition, compared to GPT models, big gaps occur in math, coding, and extraction domains. To further enhance performance, we plan to enrich our dataset with more examples in these domains.
>
> Table R1. Evaluation results of our proposed and baseline chat models in different domains on MT-Bench. We show the average scores in Overall column. Among those open source models, we make the best performance in bold and the second one underlined.
> | Model           | Writing | Roleplay | Reasoning | Math | Coding | Extraction | Stem | Humanities | Overall |
> |-----------------|:-------:|:--------:|:---------:|:----:|:------:|:----------:|:----:|:----------:|:-------:|
> | GPT-3.5-Turbo   |   $9.20$  |   $8.40$   |    $5.65$   | $6.30$ |  $6.90$  |    $8.85$    | $8.70$ |    $9.55$    |   $7.94$  |
> | GPT-4           |   $9.65$  |   $8.90$   |    $9.00$   | $6.80$ |  $8.55$  |    $9.38$    | $9.70$ |    $9.95$    |   $8.99$  |
> | Baize-v2-13B    |   $7.65$  |   $6.80$   |    $5.40$   | $1.80$ |  $3.00$  |    $4.60$    | $7.73$ |    $9.03$    |   $5.75$  |
> | UltraLM-13B     |   $8.15$  |$\underline{7.25}$|    $4.75$   | $2.60$ |$\underline{3.45}$|$5.55$    | $8.00$ |$\underline{9.55}$|   $6.16$  |
> | Vicuna-v1.3-13B | $\bf{9.25}$|   $7.18$   |  $\bf{5.85}$ | $2.60$ |  $3.25$  |    $5.55$    | $7.98$ |    $9.45$    |   $6.38$  |
> | Vicuna-v1.5-13B |   $8.10$  |   $8.05$   |   $4.15$ |$\underline{3.45}$|  $3.00$  |$\underline{6.35}$|$\underline{8.50}$|9.45 |$\underline{6.57}$|
> | Parrot-Chat-13B |$\underline{8.50}$| $\bf{8.40}$ |$\underline{5.50}$|$\bf{3.95}$|$\bf{3.85}$|$\bf{6.55}$  | $\bf{8.55}$|$\bf{9.65}$ | $\bf{6.81}$|

---

> ### Author Response · Authors · 2023-11-20
> **Response To Reviewer tuRa Part 2**
>
> ### **Q4: Qualitative examples.**
> We have added an appendix containing qualitive examples, including the model generated responses (**Section E**) and examples of our dataset (**Section G**). Please refer to the supplementary material.
>
> ### **Q5: Hallucinations in data.**
> Our dataset occasionally exhibits instances of "hallucinations," where ChatGPT generates responses that contain information not aligned with reality. This phenomenon is somehow  unavoidable for now, even in existing datasets such as ShareGPT and UltraChat.
>
> More importantly, we have observed that even when ChatGPT generates hallucinatory responses, our Parrot-Ask model can identify mistakes and guide ChatGPT to produce correct responses in subsequent interactions. For example, when ChatGPT initially incorrectly stated that automating the Cisco AnyConnect Secure Mobile Client via a script was not possible, our Parrot-Ask model pointed out that it could be accomplished using C#.  ChatGPT then revised its initial response and provided relevant examples, successfully correcting a hallucination. We have shown more such examples like this in **Appendix F**.
>
> We find this discovery intriguing, as it suggests that our asking model may have the potential to correct model hallucinations through multi-turn interactions. This is a topic we intend to explore in further detail in the future.

---

> > ### Comment · Reviewer_tuRa · 2023-11-22
> >
> > Thanks for the reply. I've read the author response and other reviews. I would like to keep my original recommendation.

---

> > > ### Author Response · Authors · 2023-11-23
> > > **Response To Official Comment by Reviewer tuRa**
> > >
> > > We are very grateful for the constructive comments you have provided during the review process. Thank you for recognizing our work. We will revise the paper in accordance with the content of the rebuttal. Once again, we sincerely appreciate the effort you have expended.

---

### Meta-Review · Area_Chair_9p2U · 2023-12-06

**Metareview:**

This paper presents an approach to create instruction tuning data for multi-turn interactions, through probing ChatGPT with Parrot-Ask to generate such multi-turn conversational data. This data is then used to train a chat model, called Parrot-Chat, which is shown to perform well according to a set of metrics. The reviewers are mainly concerned about the limited novelty in the paper, this has been raised by even the reviewer who gave the paper the highest score. While the rebuttal includes responses (i.e., explanations, and new experimentation) to questions raised by the reviewers, the limited novelty issue is not sufficiently handled.

**Justification For Why Not Higher Score:**

The paper has limited novelty, given similar previous works.

**Justification For Why Not Lower Score:**

N/A

---

### Decision · Program_Chairs · 2024-01-16

Reject